# Evaluation of a New Culture Protocol for Enhancing Fungal Detection Rates in Respiratory Samples of Cystic Fibrosis Patients

**DOI:** 10.3390/jof6020082

**Published:** 2020-06-09

**Authors:** Tobias G.P. Engel, Marlou Tehupeiory-Kooreman, Willem J.G. Melchers, Monique H. Reijers, Peter Merkus, Paul E. Verweij

**Affiliations:** 1Department of Medical Microbiology, Radboud University Medical Center, 6525 GA Nijmegen, The Netherlands; Marlou.Tehupeiory-Kooreman@radboudumc.nl (M.T.-K.); Willem.Melchers@radboudumc.nl (W.J.G.M.); Paul.Verweij@radboudumc.nl (P.E.V.); 2Center of Expertise in Mycology Radboudumc/Canisius Wilhelmina Ziekenhuis, 6525 GA Nijmegen, The Netherlands; Monique.Reijers@radboudumc.nl; 3Department of Pulmonology, Radboud University Medical Center, 6525 GA Nijmegen, The Netherlands; 4Department of Pediatrics, Radboud University Medical Center, 6525 GA Nijmegen, The Netherlands; Peter.Merkus@radboudumc.nl

**Keywords:** cystic fibrosis, fungi, aspergillus, culture protocol, DG18, epidemiology

## Abstract

Cystic fibrosis (CF) can be complicated by fungal infection of the respiratory tract. Fungal detection rates in CF sputa are highly dependent on the culture protocol and incubation conditions and thus may lead to an underestimation of the true prevalence of fungal colonization. We conducted a prospective study to evaluate the additional value of mucolytic pre-treatment, increased inoculum (100 μL), additional fungal culture media (Sabouraud agar; SAB, Medium B+, *Scedosporium* selective agar; SceSel+ and Dichloran-Glycerol agar; DG18) and longer incubation time (3 weeks) compared with our current protocol. Using the new protocol, we prospectively analyzed 216 expectorated sputum samples from adult and pediatric CF patients (*n* = 77) and compared the culture yield to a three year retrospective cohort that used direct 10 μL loop inoculation on SAB with 5 days incubation (867 sputum samples/103 patients). Detection rates for molds increased from 42% to 76% (*p* < 0.0001). Twenty-six percent of cultures were polymicrobial in the prospective cohort as opposed to 4.7% in the retrospective cohort (*p* < 0.0001). Colonization rate with *A. fumigatus* increased from 36% to 57%. SAB and DG18 showed the highest detection rates for all molds (SAB 58.6%; DG18 56.9%) and DG18 had the best performance for molds other than *A. fumigatus*. The larger sample volume and longer incubation also contributed to the increased recovery of molds. The introduction of a modified fungal culture protocol leads to a major increase in detection rate and the diversity of molds, which influences fungal epidemiology and may have implications for treatment decisions.

## 1. Introduction

Cystic fibrosis (CF) is a life-limiting, hereditary, multi-organ disease and is most prevalent in Australia, Europe and North America [1]. CF is caused by a mutation in the cystic fibrosis transmembrane conductance regulator (*CFTR*) gene leading to mucus retention in the lungs, which, through repeated infection, causes progressive and irreversible damage to the lungs [1]. Whereas most infections are caused by bacteria like *Staphylococcus aureus* and *Pseudomonas aeruginosa*, studies show an increase of fungi isolated from CF respiratory secretions [1,2]. An interaction between bacterial pathogens and filamentous fungi is likely, but whether this is antagonistic of symbiotic remains unclear. Pyocyanin, a toxine produced by *P. aeruginosa*, inhibits the growth of *A. fumigatus* in vitro [3]. Conflictingly, proteases produced by *P. aeruginosa* have been suggested to promote *A. fumigatus* sensitization [4]. At present, there is no clear understanding of the clinical significance of fungal colonization and whether treatment should be advised but high-quality fungal diagnostics are important as it influences both patient care and epidemiology [5,6].

There is great variation in the reported prevalence of fungal colonization, which is partly explained by the dissimilarity of culture protocols between different health care facilities [7]. Central factors influencing the detection rate of fungi are used culture media, pre-treatment procedures, quantity of inoculated sputum and incubation time.

Hong et al. reported that only one-quarter of clinically significant fungal organisms were successfully detected with standard bacterial culture medium and selective fungal culture media provided significantly better rates of detection [8]. However, there is great variability in fungal culture media and their performance. Sabouraud (SAB) agar represents the most used mycological culture medium and facilitates the growth of the majority of filamentous fungi. Specific media have been developed with the goal to further increase fungal detection rates. These media often prevent bacterial and/or fungal overgrowth or are enriched with certain nutrients to increase the yield of specific fungi [9,10,11,12,13,14,15,16].

Masoud et al. [17] showed that their culture method with the homogenization of CF sputum with dithiothreitol (DTT) increased the sensitivity for fungal detection significantly. Conventional laboratory analysis without the homogenization of sputum samples failed to detect fungi in 24% of the included patients. Furthermore, there was an increased colony forming unit (CFU) count in 69% of the pre-treated samples [17]. It was hypothesized that chemical homogenization with DTT facilitates liquefaction of highly viscous CF sputa and leads to homogenous distribution of fungi [17].

The importance of the quantity of the inoculum in sputum samples of COPD patients was presented by Pashley et al. [18] In their experiment with 55 sputum samples 44% of *A. fumigatus* positive samples was detected while using 100 µL of homogenized sputum opposed to 19% when using 10 μL [18]. Additionally, high-volume culture also showed a significantly enhanced culture rate in non-CF patients [19,20].

Lastly, a sufficient incubation period is of importance. Whereas faster growing fungi species, such as *Aspergillus* and *Penicillium*, grow within one week, slow-growing species, such as *Scedosporium* and *Exophiala*, may require a longer incubation time [10,14]. In the experiment of Morris et al., 81% of molds were recovered by day 7 and more than 96% by day 14 [21].

The objective of our current study was to evaluate the performance of a new culture protocol, incorporating mucolytic pre-treatment, a larger inoculum, the addition of culture media and longer incubation time, and to compare its performance to our current culture protocol for sputum samples from CF patients. Furthermore, we aimed to identify the additional value of each intervention and individual culture medium.

## 2. Materials and Methods

### 2.1. Study Design

This was a 12-month prospective study involving the collection and analysis of expectorated sputa from adult and pediatric CF patients at the Radboud University Medical Center in Nijmegen, the Netherlands. Consecutive expectorated sputum samples were collected during routine quarterly outpatient visits and in case of clinical deterioration and/or hospitalization between October 2018 and October 2019. The results of the altered culture protocol of the prospective cohort were compared to a retrospective cohort of the preceding 3 years (October 2015 until October 2018). No additional samples were taken for the benefit of this study and sputum collection was identical for the prospective and retrospective cohort. Before initiation, the study protocol was discussed with the pulmonology and pediatrics department. It was agreed that only fungal growth on the traditional culture media (SAB and/or bacterial media) would be communicated. Upon request, the fungal growth on the additional fungal media was shared. The diagnosis of CF disease was based on typical clinical characteristics alongside a positive sweat test (chloride > 60 mmol/L) and/or the presence of two known pathogenic CFTR mutations.

### 2.2. Mycological Analysis

#### 2.2.1. The Following Standardized Protocols Were Used

Retrospective cohort (mycological analysis): Direct loop inoculation of an unstandardized amount (±10 μL) of sputum on SAB agar with chloramphenicol and gentamicin 28 °C (Becton Dickinson, Franklin Lakes, NJ, USA). All plates were incubated for 5 days and were evaluated daily for the presence of fungal growth.

Prospective cohort (mycological analysis): Before the inoculation of fungal culture media, sputa were homogenized for 30 min at 37 °C with DTT (Sigma, Saint Louis, MO, USA) in a 1:2 (*v*:*v*) ratio. After incubation, 100 μL of digested sputum was inoculated onto each of the 4 semi-selective culture media, SAB 28 °C and Medium B+ 28 °C, *Scedosporium* selective agar (SceSel+) 37 °C, Dichloran-Glycerol Agar with chloramphenicol and gentamicin (DG18) 28 °C (all 3 homemade), and subsequently incubated aerobically for 3 weeks with daily evaluation for growth of molds in week 1 and biweekly evaluation in week 2 and 3. Volume-limited samples were cultured in a fixed order; 1. SAB 2. SceSel+ 3. Medium B+ 4. DG18. The remaining sputa were stored at −80 °C. All components of the homemade fungal media and the performed quality control measurements can be found in the Appendix A.

Both cohorts: Direct loop inoculation of an unstandardized amount (±10 μL) of sputum on bacterial culture media (Columbia III Agar with 5% Sheep Blood 36 °C CO_2_, Chocolate Agar with IsoVitaleXtm and Bacitracine 36 °C anaerobe, MacConkey agar 36 °C O_2_ (Becton Dickinson, Franklin Lakes, NJ, USA), *Burkholderia cepacia* selective Agar 36 °C O_2_ and Chapman agar 36 °C O_2_ (Oxoid, Basingstoke, UK). All plates were incubated for 5 days and were evaluated daily for the presence of fungal (and bacterial) growth. The excessive growth of yeasts was reported in both cohorts but not evaluated in this study. Molds were identified by their macroscopic and microscopic morphology. A visual overview of the study protocol can be found in Figure 1.

#### 2.2.2. Rationale of the Selected Fungal Culture Media

SAB medium was maintained as it represents the usual mycological culture medium and facilitates growth of most filamentous fungi. Medium B+ also enables growth of the majority of fungi but contains a larger number of antibiotics (ceftazidime, chloramphenicol, colistin and cotrimoxazole). Nagano et al. showed that adding these antibiotics to fungal culture media increased the yield of fungi in their experiments and hypothesize this was the result of less bacterial overgrowth and bacterial toxins like pyocyanine [9]. We included Medium B+ to potentially replace SAB. *Scedosporium* selective agar was added for its property to inhibit the growth of fast-growing fungi, such as *Aspergillus* and *Penicillium* spp., and therefore prevent the overgrowth of slower growing fungi, such as *Scedosporium* spp., that are benomyl resistant [12]. Multiple studies show an increased isolation rate of *Scedosporium* spp. in clinical samples with the use of SceSel+ [10,11,13]. Dichloran-Glycerol Agar is a semi-selective medium used mainly in environmental research and no previous application in clinical samples [22]. Due to the properties of this dichloran-containing medium, fungal colonies remain small. We hypothesized that the isolation rate of slower growing and/or less abundant fungi would increase due to prevention of fungal overgrowth. To lessen bacterial overgrowth we added chloramphenicol and gentamicin to this media.

Unfortunately no commercial preparations are available for Medium B+ and SceSel+, and they were prepared in our center according to the method of Nagano et al. [9] and Rainer et al. [12], respectively. The preparation of DG18 was the least laborious, as DG18 agar base is commercially available and can easily be produced in-house.

### 2.3. Primary and Secondary Outcomes

The primary outcome of this study was the difference in the detection rate of molds between the retrospective and prospective cohort. The detection rate was defined as the percentage of cultures with growth of a (specific) mold.

Diversity was assessed in both cohorts and defined as both the total amount of unique molds isolated in one year as the amount of polymicrobial cultures (≥2 molds per sputum culture).

Furthermore, colonization with *A. fumigatus* was determined. Colonization was defined as the recovery of *A. fumigatus* from ≥50% of sputum samples in patients with a minimum of 3 or more sputum samples per year.

This study was not intended as an epidemiology study; however, prevalence could be calculated. Culture was considered positive for a given species and a given sputum sample when the species growth was evidenced on at least one of the fungal or bacterial culture media.

In the prospective cohort, sub-analyzes were performed. The time until detection was reported for each fungal culture medium and fungus as the first week of positive culture. Furthermore, the additional value of each intervention (1: sample pre-treatment and larger inoculum, 2: prolonged incubation, 3: additional fungal culture media) on the detection rate of molds was assessed.

The detection rate, positive percent agreement and *A. fumigatus* colonization rate of each semi-selective fungal medium was determined. From this analysis, we excluded samples who only had growth on bacterial culture media, in which not all four fungal media were inoculated (e.g., volume-limited sample) or in which growth of molds could not be traced back to a specific medium (e.g., technician failed to register properly). The positive percent agreement was defined as the proportion of positive cultures for (specific) molds for the respective culture medium over the total number of positive cultures using all 4 media.

In all analysis where fungal groups are reported *Scedosporium* species also include *Lomentospora prolificans*.

### 2.4. Statistical Analysis

The detection rate, proportion of polymicrobial cultures, *A. fumigatus* colonization, time until detection and the positive percent agreement were compared between the different cohorts, years of the study, and culture media using Fishers exact test. We used *t*-tests and one-way ANOVA to assess potential differences in demographic variables and/or the amount of sputum cultures per patient in the several years of the study.

## 3. Results

### 3.1. Study Population

In the prospective cohort, 261 samples from 77 patients were collected. In the retrospective cohort, 867 samples (year 1, 310 samples; year 2, 303; year 3, 254) were collected from 103 unique patients (year 1 75 patients; year 2 81 patients; year 3 84 patients). Seventy-five of the prospectively included patients had previously been sampled in the retrospective cohort (year 1, 60 patients; year 2, 67 patients; year 3, 70 patients). The mean age of the entire population was 29.3 years and 49.5% was female. A median of three sputum cultures per patient were taken. Age, sex and the number of sputum cultures did not differ significantly between the different years of the study or between the retrospective and prospective cohort.

### 3.2. Retrospective Cohort vs. Prospective Cohort

Using the new culture protocol, seventy-six percent of all sputum cultures were positive with a mold in the prospective cohort, presenting a significant increase (*p* < 0.0001 for all years) when compared to the rate of detection in the retrospective cohort (42%). The increase in the rate of detection between the prospective and retrospective cohort reached significance for *A. fumigatus* (*p* < 0.0001)*, Aspergillus* species (*p* < 0.0001)*, Penicillium* species (*p* < 0.0001) and *Exophiala* species (*p* = 0.011). Significance was also reached for “Other fungi” (all fungi except *Aspergillus*, *Penicillium*, *Scedosporium*/*Lomentospora* and *Exophiala*) (*p* < 0.0001) and “Any mold excluding *A. fumigatus”* (*p* < 0.0001). The detection rates of the most prevalent molds are summarized in Table 1. An overview of all cultured fungi and prevalence data can be found in the Appendix A. Appendix A also shows that the variation in detection rate for the most prevalent molds was low in the three years of the retrospective cohort. Similar detection rates were observed when only patients included in both the retrospective and prospective cohort were analyzed (Appendix A).

Diversity was higher in the prospective cohort compared to the retrospective cohort, with both an increase in the total amount of unique molds recovered and the number of polymicrobial cultures (≥2 molds per sputum culture). In the prospective cohort, 24 different fungal species (excluding unidentified fungi) were cultured versus, respectively, 10 (year 1 and 2) and 11 (year 3) unique species in the retrospective cohort. Twenty-six percent of cultures were polymicrobial in the prospective cohort compared with 4.7% in the retrospective cohort (*p* < 0.0001), with two cultures showing up to five unique species. The most frequent associations were *A. fumigatus* with, respectively, *Penicillium* spp. (*n* = 25), *A. flavus* (*n* = 6) and *A. niger* (*n* = 4). 

A total of 51 patients in the prospective cohort had three or more sputum samples taken. Of these patients, more than half (56.9%) were considered to be colonized with *A. fumigatus (*≥50% of samples positive with *A. fumigatus*). This was a significant increase when compared to the retrospective cohort (35.8%; *p* = 0.009) (Table 1).

### 3.3. Effect of Each Protocol Modification on Fungal Detection Rates

The detection rate of molds in the retrospective cohort was 42% compared to 55.6% in the prospective cohort when analyzed using comparable culture conditions (SAB after one week of incubation; *p* = 0.0001). Therefore, as variation in detection rate was low in the three years of the retrospective cohort, this difference in recovery rate can most likely be attributed to the mucolytic pre-treatment and greater inoculation volume of the samples. After three weeks of incubation of SAB plates, the detection rate increased only to 62% (*p* = 0.1547). With the addition of DG18, the detection rate further increased to 72.8% (*p* = 0.0116). The additional value of SceSel+ and Medium B+ was minor (75.8%; *p* = 0.4831) (Figure 2).

### 3.4. Time until Detection

The majority of molds (74.3%) grew within the first week of incubation, while 19.6% were found in the second week and 6.2% in the third week. Growth on Medium B+ was significantly slower (45.2% after the first week of incubation) when compared to SAB (14.6% *p* < 0.0001) and DG18 (22.4% *p* < 0.0001). Even on the medium that facilitates the fastest growth (SAB), 8.4% of *A. fumigatus* were missed with only 1 week of incubation. This increased to 27.4% when “any mold excluding *A. fumigatus*” were evaluated. This was mainly because of positive cultures in the second week. On SAB and DG18, respectively, 98.5% and 93.2% of *A. fumigatus* and 96.8% and 91.5% of “any mold excluding *A. fumigatus*” grew in the first 2 weeks. The time until detection was also dependent on the fungal genus/species. For instance, 39.5% of the “other fungi” and 100% of the Basidiomycetes grew only after week 1. A more detailed overview of time to detection for different fungi and culture media can be found in the Appendix A.

### 3.5. Individual Performances of Fungal Culture Media

SAB and DG18 showed the highest detection rate for molds (SAB 58.6%; DG18 56.9%), which was more when compared to Medium B+ (47.8%; SAB *p* = 0.026; DG18 *p* = 0.063). The detection rate for “any mold excluding *A. fumigatus*” was higher for DG18 (29.3%) when compared to SAB (22.4%; *p* = 0.1115) and Medium B+ (16.4%; *p* = 0.0013). The difference in the total number of “non *A. fumigatus”* isolates was notable (DG18 *n* = 80; SAB *n* = 55; Medium B+ *n* = 43), with up to four “non *A. fumigatus* isolates” per DG18 culture. SceSel+ had only six positive cultures in the whole cohort (*A. fumigatus n* = 2; *L. prolificans n* = 2; *Rhizomucor* spp. *n* = 2). One *L. prolificans* was only cultured on the SceSel+, but a *S. apiospermum* was missed and grew only on Medium B+. The detection rate of the individual culture media for the most prevalent molds are summarized in Table 2. The positive percent agreement of the individual culture media with all media can be found in Table 3.

## 4. Discussion

This study shows that a modified fungal culture protocol can significantly increase the detection rate for molds in sputum samples of CF patients. This effect was observed for the total spectrum of molds and led to both an increase of patients colonized with the frequently cultured *A. fumigatus* but also to an increase in the detection rate of other fungi.

Analyses of individual components of the new protocol showed that mucolytic pre-treatment combined with a higher inoculum resulted in a 14% increase in detection rate when compared to the retrospective protocol. Although we did not directly compare both protocols prospectively, it is most likely that the observed increase was indeed due to these modifications and not caused by changes within subjects or exposure. This is supported by the low variation in detection rate in the three years prior to the prospective cohort (retrospective cohort). Furthermore, a subanalysis of patients that were included in both the retrospective and prospective cohort showed a similar increase in detection rate (Appendix A).

When fungal growth in week 2 and 3 was included in the analysis, the detection rate of the prospective cohort using SAB increased an additional 6%, which was not significant. However, many SAB-plates were overgrown with *A. fumigatus* during prolonged incubation, which hampers the detection of slower growing fungi. The prolongation of incubation did result in a significant increase in detection rate when the combination of SAB and DG18 was evaluated after week 1 and week 3 (increase of 10%, *p* = 0.0242).

The fungal detection rate was significantly higher when DG18-plate was added (73% SAB + DG18 in week 3 compared with 63% with SAB alone, *p* = 0.0116), which underscores the benefit of adding DG18. The further addition of SceSel+ and Medium B+ only marginally improved detection rate. Although SAB and DG18 had the best individual performance, their combination was not significantly better than a combination of either medium with Medium B+.

This is the first study that used DG18 on human diagnostic samples. Through its potential to allow for the growth of most fungi but simultaneously restrict the colony diameters of fast-growing fungi, it prevented overgrowth and proved to be the medium with the highest performance for molds other than *A. fumigatus*. Based on our findings, we recommend adding DG18 to the fungal culture protocol to culture these more rare and slow-growing fungi. Another advantage of DG18 opposed to SceSel+ and Medium B+ is that DG18 agar base is commercially available.

SceSel+ adequately suppressed the growth of the faster growing molds, but the detection rates of *Scedosporium* spp. did not increase. Low recovery rates of *Scedosporium* spp. may be due to the low prevalence of patients colonized with *Scedosporium* spp. in our center. Over the whole study period, only three out of 105 patients had a *Scedosporium* spp. positive culture with only one patient with consecutive positive cultures. SceSel+ might still be of additional value in a setting with more *Scedosporium* spp. as is suggested by previous publications [10,11,13]. Modifications of SceSel+ have been proposed, but have not yet been evaluated on clinical samples [23].

In contrast to the findings of Nagano et al. the performance of Medium B+ was inferior when compared to SAB in this study. A possible explanation is that we used SAB supplemented with other antibiotics than in their experiment (chloramphenicol and gentamicin vs. chloramphenicol, cotrimethoxazole, ceftazidime and colistin) and a different incubation temperature (28 vs. 22 °C) [9].

Our improved diagnostic protocol did not include a specific culture medium for the isolation of *Exophiala* spp. The additional value of such media is unclear. Some authors noted that it is beneficial for *E. dermatitidis* detection [15], but its advantage when directly compared to SAB with gentamycin and chloramphenicol has not yet been established [16]. A recent French study indicated that the sensitivity of erythritol enriched medium dropped threefold when the media were incubated at 37 °C compared to 20–27 °C. Unfortunately, the low numbers of cultured *Exophiala* isolates did not allow them to draw any conclusions about the medium’s performance [14].

The clinical significance of fungi cultured from respiratory secretions of CF patients is still under debate [5,6]. *A*. *fumigatus* and *S. apiospermum* have been proposed as “Category 1 fungi” as they have been widely reported as etiological agents of invasive disease or associated with Allergic Bronchopulmonary Mycoses (ABPM) [24]. Additionally, lower respiratory infections with *Aspergillus* species are associated with the progression of structural lung disease in young children with CF [25]. However, the pathogenicity of other fungi has also been reported [26,27,28,29]. Until it is clear which fungi are clinically relevant, clinicians and microbiologists have to decide together to what extent and which fungi they wish to culture. 

Criteria to distinguish the mere presence of fungal spores from colonization and infection have been proposed. The proposed microbiological criterium involves positive cultures with the same fungus in >50% [5] or two or more respiratory samples in the last 6–12 months [30]. Our center patients with pulmonary deterioration colonized with fungi who do not improve under broad-spectrum antibiotics are evaluated for antifungal treatment. As the colonization rate of *A. fumigatus* increased from 36% to 57% with our altered culture protocol, this might lead to an increase in the number of treated patients.

This study shows that the introduction of an altered fungal culture protocol, including mucolytic pre-treatment, a higher inoculum, the addition of specific culture media and prolonged incubation, can lead to a major increase in the detection rate and diversity of molds. Improved culture protocols will help to better understand the role of fungi in CF.

## Figures and Tables

**Figure 1 jof-06-00082-f001:**
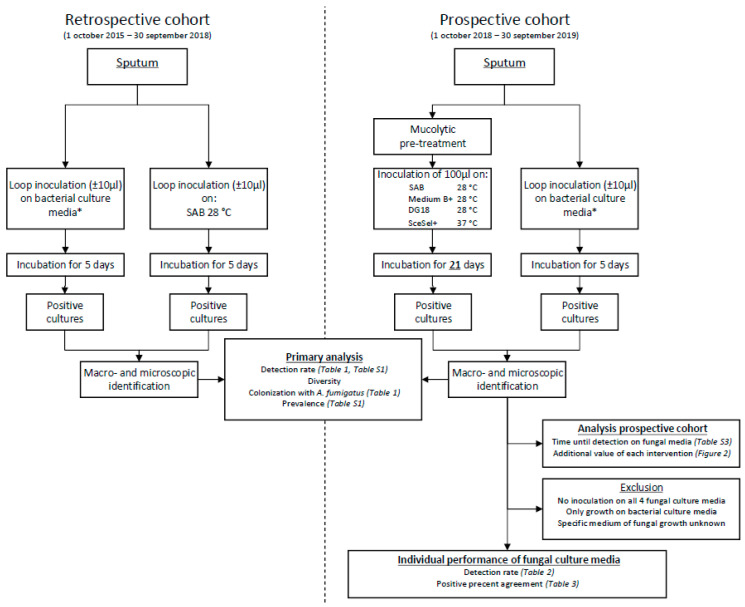
Flow diagram depicting the study design. SAB, Sabouraud; DG18, Dichloran-Glycerol Agar; SceSel+, *Scedosporium* selective agar. * Bacterial culture media: Columbia III Agar with 5% Sheep Blood 36 °C CO_2_, Chocolate Agar with IsoVitaleXtm and Bacitracine 36 °C anaerobe, MacConkey agar 36 °C O_2_, Burkholderia cepacia selective Agar 36 °C O_2_ and Chapman agar 36 °C O_2_.

**Figure 2 jof-06-00082-f002:**
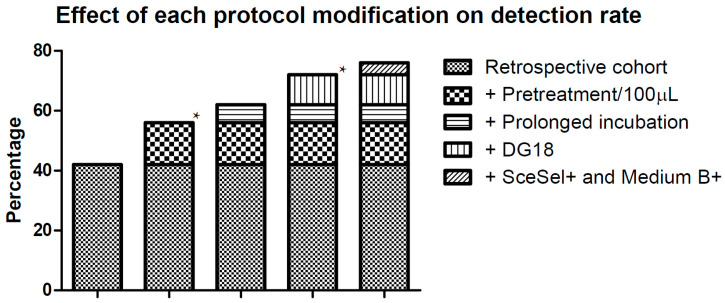
Effect of each protocol modification on fungal detection rates. Retrospective cohort; the detection rate of any mold of the retrospective cohort + Pretreatment/100 μL; the detection rate of prospective cohort after 1 week of incubation on Sabouraud (SAB) with mucolytic pretreatment and inoculation of 100 μL sputa + Prolonged incubation; the detection rate after 3 weeks of incubation on SAB with mucolytic pretreatment and the inoculation of 100 μL sputa + DG18; the detection rate after 3 weeks of incubation on SAB and Dichloran-Glycerol Agar (DG18) with mucolytic pretreatment and inoculation of 100 μL sputa + SceSel+ and Medium B+; the detection rate after 3 weeks of incubation on SAB, DG18, *Scedosporium* selective agar (SceSel+) and medium B+ with mucolytic pretreatment and inoculation of 100 μL sputa. * Significant increase compared to the previous column.

**Table 1 jof-06-00082-t001:** Detection rate of molds in the sputum samples of cystic fibrosis (CF) patients.

Type of Mold	Rate of Detection	
	Retrospective Cohort (*n* = 867)	Prospective Cohort (*n* = 261)
Any mold	42%	75.9% *
*A*. *fumigatus*	33.8%	55.9% *
*Aspergillus* species	4.3%	12.6% *
*Penicillium* species	5.1%	23% *
*Exophiala* species	0.1%	1.5% *
*Scedosporium* species ^a^	0.9%	1.1%
Other molds ^b^	2.5%	14.9% *
Any mold excluding *A*. *fumigatus*	12%	41.8% *
*A*. *fumigatus* colonization (prevalence) ^c^	35.8%	56.9% *

^a^ Includes both *Scedosporium apiospermum* and *Lomentospora prolificans.*
^b^ All molds that do not group in any of the above. ^c^ Defined as recovery of *A. fumigatus* from >50% of sputum samples in patients with a minimum of 3 or more sputum samples per year. * Significant increase (*p* < 0.05) between the prospective cohort and the retrospective cohort.

**Table 2 jof-06-00082-t002:** Detection rate of individual culture media for molds.

Type of Mold	Rate of Detection					
	SAB ^a^	Medium B+	DG18 ^a^	SceSel+ ^a^	All Media	SAB/DG18 ^a^
Any mold	58.6%	47.8%	56.9%	2.6%	73.7%	69.8%
*A. fumigatus*	44%	35.8%	37.5%	0.9%	53.9%	50%
*Aspergillus* species	4.3%	4.3%	6.9%	0%	11.6%	9.5%
*Penicillium* species	12.5%	8.2%	19.4%	0%	24.6%	22.8%
*Exophiala* species	0.9%	0.4%	0.4%	0%	1.3%	1.3%
*Scedosporium* species ^b^	0.4%	0.4%	0.4%	0.9%	1.3%	0.4%
Other molds ^c^	5.6%	4.7%	6.0%	0.9%	14.2%	10.8%
Any mold excluding *A.fumigatus*	22.4%	16.4%	29.3%	1.7%	42.2%	37.1%
*Aspergillus fumigatus* colonization (prevalence) ^d^	50%	41.3%	41.3%	0%	56.5%	52.2%

^a^ SAB, Sabouraud; DG18, Dichloran-Glycerol Agar; SceSel+, *Scedosporium* selective agar. ^b^ Includes both *Scedosporium apiospermum* and *Lomentospora prolificans.*
^c^ All molds that do not group in any of the above. ^d^ Defined as recovery of *A. fumigatus* from >50% of sputum samples in patients with a minimum of 3 or more sputum samples per year.

**Table 3 jof-06-00082-t003:** Positive percent agreement of individual culture media with all media.

Type of Mold	Positive Percent Agreement			
	SAB ^a^	Medium B+	DG18 ^a^	SAB/DG18 ^a^
Any mold	79.5% (CI95% 72.7–85.3)	64.9% (CI95% 57.3–72.0)	77.2% (CI95% 70.1–83.3)	94.7% (CI95% 90.2–97.6)
*A. fumigatus*	81.6% (CI95% 73.7–88)	66.4% (CI95% 57.4–74.6)	69.6% (CI95% 60.7–77.5)	92.8% (CI95% 86.8–96.7)
Any mold excluding *A. fumigatus*	53.1% (CI95% 42.7–63.2)	38.8% (CI95% 29.1–49.2)	69.4% (CI95% 59.3–78.3)	87.8% (CI95% 79.6–93.5)

^a^ SAB, Sabouraud; DG18, Dichloran-Glycerol Agar.

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
