# Peer review of "Evaluation of a New Culture Protocol for Enhancing Fungal Detection Rates in Respiratory Samples of Cystic Fibrosis Patients"

_jof, 2020, doi:10.3390/jof6020082_

Round 1

Reviewer 1 Report

Lung infections in cystic fibrosis patients predominantly influence morbidity and mortality. In addition to bacteria, the lungs of CF patients are often infected by fungi. The manuscript jof-828364 presents a suggestion of a new laboratory protocol for a better recovery of fungal isolates from sputum of cystic fibrosis patients, aiming to achieve a better diagnosis of possible fungal infection/colonization.

This is an interesting paper, with possible direct impact on diagnosis. Some suggestions are made:

Introduction: Given the importance of bacteria and fungi in respiratory products, I suggest the introduction of couple of sentences highlighting the role and interaction of fungi and bacteria in cystic fibrosis: antagonistic or symbionts?

Lines 61-64: Other references regarding high volume cultures should be mentioned:

https://www.ncbi.nlm.nih.gov/pmc/articles/PMC5631283/

https://www.sciencedirect.com/science/article/abs/pii/S1198743X19306196

Methods:

The major limitation of this study is the comparison between data obtained retrospectively and prospectively, in different conditions. The comparative study should be in the same samples.

Why the authors did not perform, in the prospective study, some assays in parallel, at least regarding the pretreatment and volume inoculated?

The authors inoculate the sputum samples in plates but one thing to consider may be the use of tubes with agar slants, given the proposed extended period of incubation.

Please put in italics in species/genera names (Scedosporium, for ex, line 98, 125)

Line 143: The authors refer polymicrobial cultures…which were the most frequent associations?

Results:

Table 1,2: “other fungi” does it includes yeasts? That should be mentioned.

Line 234-237: There is no need to describe in detail the samples excluded, since those criteria were already described in lines 154-158.

Line 243-244: “….cultures in the whole cohort (A. fumigatus n=2; L. prolificans n=2; Rhizomucor spp. n= 2). One L. prolificans was only cultured on the SceSel+, but a S. apiospermum was missed.” Where did this Scedosporium grew?

I suggest the introduction of a figure with the proposed algorithm of the proposed laboratory method for CF patients’ sputum samples.

Author Response

Response to reviewer 1 comments

Point 1: Given the importance of bacteria and fungi in respiratory products, I suggest the introduction of couple of sentences highlighting the role and interaction of fungi and bacteria in cystic fibrosis: antagonistic or symbionts?

Response: We have added the following sentence to the introduction. (Line 39-43: “An interaction between bacterial pathogens and filamentous fungi is likely, but whether this is antagonistic of symbiotic remains unclear. Pyocyanin, a toxic produced by P. aeruginosa, inhibits growth of A. fumigatus in vitro [3]. Conflictingly, proteases produced by P. aeruginosa have been suggested to promote A. fumigatus sensitization [4].”

Point 2:  Lines 61-64: Other references regarding high volume cultures should be mentioned:

https://www.ncbi.nlm.nih.gov/pmc/articles/PMC5631283/

https://www.sciencedirect.com/science/article/abs/pii/S1198743X19306196

Response:
Lines 64-65:The following sentence was added.

Additionally, high-volume culture also showed a significantly enhanced culture rate in non-CF patients [17,18].

Point 3: The major limitation of this study is the comparison between data obtained retrospectively and prospectively, in different conditions. The comparative study should be in the same samples.

Why the authors did not perform, in the prospective study, some assays in parallel, at least regarding the pretreatment and volume inoculated?

Response: We agree with the reviewer that a direct comparison from the same sample is the preferred study design. We chose a retrospective/prospective design due to the work load for our technicians, who work in routine mycology. We therefore choose to compare the new protocol with the past three years, which showed very stable culture results. In addition, we discuss the drawbacks of this study design when discussing the limitations in the discussion section.

Point 4: The authors inoculate the sputum samples in plates but one thing to consider may be the use of tubes with agar slants, given the proposed extended period of incubation.

Response: We agree with the reviewer that the extended period of incubation bears the risk that the plates become too dry. To prevent this we taped the side of the agar plates, both to prevent evaporation and to reduce the risk for contaminants. This worked well. We considered the use of agar slants, but in our experience this makes the evaluation of polymicrobial cultures more challenging.

Point 5: Please put in italics in species/genera names (Scedosporium, for ex, line 98, 125)

Response: We thank the reviewer for this observation. We have changed the species/genera names to italic.

Point 6: Line 143: The authors refer polymicrobial cultures…which were the most frequent associations?

Response: We have added the most frequent associations the results section. Line 198-199: “The most frequent associations were A. fumigatus with respectively Penicillium spp. (n=25), A. flavus (n=6) and A. niger (n=4).”

Point 7: Table 1,2: “other fungi” does it includes yeasts? That should be mentioned.

Response: We thank the reviewer for pointing out this unclarity. We mention in the methods section (line 111) that yeasts are not evaluated. To further clarify this we changed “other fungi” to “other molds” in table 1 and table 2.

Point 8: Line 234-237: There is no need to describe in detail the samples excluded, since those criteria were already described in lines 154-158.

Response: Line 234-237 were removed from the manuscript.

Point 9: Line 243-244: “….cultures in the whole cohort (A. fumigatus n=2; L. prolificans n=2; Rhizomucor spp. n= 2). One L. prolificans was only cultured on the SceSel+, but a S. apiospermum was missed.” Where did this Scedosporium grew?

Response: We changed line 245-246. “One L. prolificans was only cultured on the SceSel+, but a S. apiospermum was missed and grew only on Medium B+.”

Point 10: I suggest the introduction of a figure with the proposed algorithm of the proposed laboratory method for CF patients’ sputum samples.

Response: We thank the reviewer for this suggestion. However we believe that the laboratory method used for CF patients sputum samples should be decided locally. Until it is clear which fungi are clinically relevant, clinicians and microbiologists have to decide together to what extent and which fungi they wish to culture. Above mentioned together local epidemiology should decide the used culture media.

Reviewer 2 Report

In this manuscript, Engel and co-workers evaluated several potential improvements of a Sabouraud-based standard protocol for detection of fungi in sputa from CF patients. Detection of Aspergillus and other fungal species could be significantly improved by mucolytic pretreatment of the sputum, by plating 100 instead of 10 µl and by combining Sabouraud and Dichloran-Glycerol agar plates. This is a very interesting and helpful study. The manuscript is well written and there are only some minor points that should/could be addressed.

Specific comments:

  1. Analysis of fungi from sputum samples is hampered by the potential contamination with fungal spores that were inhaled, but remained in a resting state. The relevance of single colonies on plates is often unclear and may be due to a contamination, whereas several colonies indicate a colonization. Has this been taken into account?
  2. An incubation time of 3 weeks bears the risk that the plates may become to dry. Have the authors taken measures to avoid this?
  3. 135: … according to the methods of Nagano et al. [7] and Rainer et al. [10], respectively.
  4. 208: … plates the detection rate increased only to…
  5. Table 3: Positive percent agreement 20? Please provide an explanation for CI.
  6. Supplementary Table 1: Please explain the detection rate (n).

Author Response

Response to reviewer 2 comments

Point 1: Analysis of fungi from sputum samples is hampered by the potential contamination with fungal spores that were inhaled, but remained in a resting state. The relevance of single colonies on plates is often unclear and may be due to a contamination, whereas several colonies indicate a colonization. Has this been taken into account?

Response: We agree with the reviewer that cultures with several colonies makes colonization more likely when compared to cultures with single colonies. However, in our opinion cultures with single colonies do not rule out colonization. Therefore we reported growth independently from the amount of colony-forming-units.  

Point 2: An incubation time of 3 weeks bears the risk that the plates may become to dry. Have the authors taken measures to avoid this?

Response: We agree with the reviewer that the extended period of incubation bears the risk that the plates become to dry. To prevent this we taped the side of the agar plates. We considered the use of agar slants, but in our experience this makes the evaluation of polymicrobial cultures more challenging.

Point 3: 135: … according to the methods of Nagano et al. [7] and Rainer et al. [10], respectively.

Response: Changed according to suggestion of reviewer.

Point 4: 208: … plates the detection rate increased only to…

Response: Changed according to suggestion of reviewer.

Point 5: Table 3: Positive percent agreement 20? Please provide an explanation for CI.

Response: Thank you for this observation. The number “20”was not supposed to stand there and has been removed.

Point 6:  Supplementary Table 1: Please explain the detection rate (n).

Response: Number of fungal species recovered through culture